

# Simultaneous functioning of different light-harvesting complexes—a strategy of adaptation of purple bacterium *Rhodopseudomonas palustris* to low illumination conditions

Olga Petrovna Serdyuk[1], Azat Vadimovich Abdullatypov[1], Lidiya Dmitrievna Smolygina[1], Aleksandr Aleksandrovich Ashikhmin[1] and Maxim Alexandrovich Bolshakov[1]

Institute of Basic Biological Problems of the Russian Academy of Sciences—A Separate Subdivision of PSCBR RAS (IBBP RAS), Pushchino, Moscow Region, Russian Federation

## ABSTRACT

Novel peripheral light-harvesting (LH) complex designated as LL LH2 was isolated along with LH4 complex from *Rhodopseudomonas palustris* cells grown under low light intensity (LL). FPLC-MS/MS allowed to reveal PucABd and PucBabc apoproteins in LL LH2 complex, which is different from previously described LH4 complex containing PucABd, PucABa and PucBb. The main carotenoids in LL LH2 complex were rhodopin and 3,4-didehydrorhodopin. Three-dimensional modeling demonstrated which amino acid residues of all the $\beta$-subunits could interact with carotenoids (Car) and bacteriochlorophyll a (BChl a). Analysis of amino acid sequences of $\alpha$-subunits of both LL complexes showed presence of different C-terminal motifs, IESSVNVG in $\alpha$a subunit and IESSIKAV in $\alpha$d subunit, in the same positions of C-termini, which could reflect different retention force of LL LH2 and LH4 on hydroxyl apatite, facilitating successful isolation of these complexes. Differences of these LL complexes in protein and carotenoid composition, in efficiency of energy transfer from Car to BChl a, which is two times lower in LL LH2 than in LH4, allow to assign it to a novel type of light-harvesting complex in *Rhodopseudomonas palustris*.

## INTRODUCTION

Studies of molecular basics of formation and assembly of light-harvesting (LH) pigment-protein complexes during adaptation of phototrophic purple bacterium *Rhodopseudomonas (Rps.) palustris* to low light intensities have not led to complete understanding of this process yet (*Cogdell, Gall & Köhler, 2006*). This is due to presence of at least five Puc$_{abcde}$ genes encoding $\alpha\beta_{abcde}$-apoproteins of LH-complexes (*Evans, Hawthornthwaite & Cogdell, 1990*; *Tharia et al., 1999*; *Hartigan et al., 2002*) and complicated regulation of the expression of each gene by a certain bacteriophytochrome. The number of bacteriophytochromes is equal

Corresponding authors
Olga Petrovna Serdyuk, serdyuko@rambler.ru
Azat Vadimovich Abdullatypov, aza-tik888@yandex.ru

to the number of light-harvesting complex apoprotein genes (*Vierstra & Davis, 2000*; *Evans et al., 2005*; *Fixen, Oda & Harwood, 2019*). Proteins of LH-complexes are non-covalently bound to bacteriochlorophyll a (BChl *a*) and carotenoids (Car), and their interactions determine the ability of the complexes to absorb photons in different spectral areas and to direct the resulting energy to the reaction centers (RCs). Energy transfer is followed by its conversion into the chemical energy (*Lüer et al., 2012*). Formation of certain type of LH complex depends on illumination intensity. Under high illumination, phylogenetically close phototrophic purple bacteria *Rhodoblastus (Rbl.) acidophilus* (formerly known as *Rhodopseudomonas (Rps.) acidophila)* and *Rps. palustris* form LH2 complexes B800-850 with the main absorption maxima at 850 nm. These complexes are located peripherally in relation to core complex (LH1-RC) (*Evans, Hawthornthwaite & Cogdell, 1990*; *Gall et al., 1999*). Low illumination, or low light (LL) leads to formation of other forms of LH2 complex, LH3 (B800-820) in *Rbl. acidophilus* and LH 4 (B800) in *Rps. palustris,* with the main absorption maxima at 800 nm (*Moulisová et al., 2009*; *Kotecha, Georgiou & Papiz, 2013*; *Montemayor, Rivera & Jang, 2018*). All the types of peripheral LH complexes consist of short (50–60 amino acid residues), very hydrophobic $\alpha\beta$-polypeptides encoded by different gene pairs: PucAB$_{ab}$ for LH2, PucAB$_c$ for LH3, PucAB$_d$ for LH4 (*Taniguchi et al., 2014*). The available X-ray crystallography data (*Papiz et al., 2003*; *Brotosudarmo et al., 2011*), as well as data of single-molecule spectroscopy and molecular sieve chromatography (*Brotosudarmo et al., 2009*) showed that all the peripheral LH-complexes of these bacteria are nonameric ring-shaped structures made of symmetrically repeated basic building blocks, protomers. Each protomer is composed of one $\alpha$-subunit strongly interacting with two BChl *a* molecules and of one $\beta$-subunit bound to one monomeric BChl *a*. The BChl *a* molecules of $\alpha$-subunits form the inner ring, BChl *a* molecules of $\beta$-subunits form the outer B800 ring. Each protomer also includes one carotenoid molecule, which interacts with polypeptides and BChl *a* and stabilizes the structure of the complexes (*Lang & Hunter, 1994*; *Hashimoto, Uragami & Cogdell, 2016*; *Isaacs et al., 1995*).

These two bacteria have identical quaternary structure of LH-complexes and almost the same set of genes for the synthesis of $\alpha\beta$-chains, but they adapt differently to LL illumination conditions. When shifting from HL to LL conditions, gradual substitution of LH2 (B800-850) with LH3 (B800-820) occurs in *Rbl. acidophilus,* and the spectral changes are related to changes of two amino acids in $\alpha_{ab}$-subunits, while $\beta$-subunits remain unchanged. Amino acid residues $\alpha$ Tyr44 and $\alpha$ Trp45 of LH2 complex that bind to acetyl group of BChl *a* via hydrogen bonding, are substituted with $\alpha$ Phe44 and $\alpha$ Leu45 in LH3, that cannot form hydrogen bonds, which causes hypsochromic shift in LH3 by 30 nm. Instead of these residues, the C3-acetyl group of $\alpha$BChla B820 molecules is bound via hydrogen bond to $\alpha$ Tyr41, which is substituted in equivalent position of HL LH2 with Phe (*Taniguchi et al., 2014*).

Quite a different situation is observed in LL conditions in *Rps. palustris*. In this bacterium, LH4 complex is formed of heterogenic polypeptides. Apart from $\alpha\beta_d$ subunits specific for LH4 complex (B800), it includes also polypeptides of LH2 complex, $\alpha\beta_a$ and $\beta_b$. Hence, there is always an absorption maximum at 850 nm in LH4 complex in addition to 800 nm, and the 800 to 850 nm ratio increases upon gradual decrease of illumination from

200 to 10 lx (*Brotosudarmo et al., 2011*). Absence of absorption maximum at 850 nm in LH4 complex was demonstrated only for a *Rps. palustris* mutant strain with four gene pair deletions, that had only PucD genes of all the genes of biosynthesis of light-harvesting complexes (*Southall et al., 2018*).

The data on gradual changes of polypeptide composition of LH4 complex in this range of illumination intensity (200-10 lx) are absent in the literature.

Another complex was isolated from *Rps. palustris* under illumination conditions of 50 lx, along with the LH4 complex, and it was designated as LL LH2 (*Serdyuk, Smolygina & Ashikhmin, 2020*). It differed from the LH4 and LH2 complexes of this bacterium by the absorbance in the carotenoid region of the absorption spectrum (400–550 nm), by the ratio of maxima at 800/850 nm, and by the different retention strength of the complexes on hydroxylapatite (HA), with which the complexes were isolated. However, the LH4 and LL LH2 complexes slightly differed in the composition of carotenoids determined by the HPLC method and in the efficiency of energy transfer from Car to BChl *a*. The obtained data did not give an idea of the reason of significant differences between the two types of LL complexes, which required new experiments.

The aim of this work was to confirm the earlier suggestion about LL LH2 as a new type of complex (*Serdyuk, Smolygina & Ashikhmin, 2020*), which is formed along with LH4. The main task was to study the protein composition of LL LH2 as the main factor determining the type of complex (*Tadros & Waterkamp, 1989*; *Papiz et al., 2003*; *Brotosudarmo et al., 2011*). For this purpose, a minimum illumination of 10 lux was used, at which the ratio of 800/850 nm in the absorption spectrum of the LH4 complex was still unchanged (*Brotosudarmo et al., 2011*); the LL LH2 complex obtained on HA was additionally purified by HPLC on columns with a weak ion exchanger DEAE-TOYOPEARL 650 S and mild detergent dodecyl maltoside (DDM), which did not change its protein content. Other goals were to study the composition of Car, to calculate the efficiency of energy transfer from Car to BChl *a*, and to determine of the sites of interaction of $\beta$-subunits with pigments on three-dimensional models.

## MATERIAL AND METHODS

The object of the study was purple phototrophic non-sulfur bacterium *Rps. palustris*, strain KM MGU 286 (le5 strain, collection of Microbiology Chair of Freiburg University, Germany (Lehrstuhl fur Mikrobiologie, Freiburg, BRD)), that was kindly granted by Microbiology chair of Lomonosov Moscow State University. The bacteria were cultivated anaerobically on Hutner's medium (*Hutner, 1946*) with ammonia succinate as nitrogen source at pH 6.8, illumination 10 lx and $t = 30\,°C$. *Rps. palustris* cells were grown in glass flat vessels 0.5 L in volume, two cm thick, for 16 days. The vials were turned around twice a day for uniform illumination. After the onset of the stationary growth phase with a constant cell density $OD_{850} = 0.7\ cm^{-1}$, the cells were precipitated by centrifugation at $10,000 \times g$, washed with 10 mM Tris–HCl buffer solution and precipitated again. The cells were then resuspended in 10 mM buffer at a ratio of 1:8 (weight/volume), the thin glass with cells was placed on ice and sonicated at 22 kHz, 8 times for 90 s at 3 min intervals to

prevent overheating. After sedimentation of the cell debris, chromatophores were isolated from the supernatant on ultracentrifuge at 144,000 g. The chromatophores were incubated in 20 mM Tris–HCl solution with 2% Triton X-100 for 40 min and placed onto a column with hydroxyapatite (HA) that was prepared according to the method used in earlier works (*Siegelman, Wieczorek & Turner, 1965*; *Serdyuk, Smolygina & Khristin, 2018*) and equilibrated with 10 mM Tris–HCl buffer solution, pH 7.8. Then the column was washed with the same buffer solution to eliminate the residual detergent (Triton X-100), and to elute the compounds that were bound weaker than the LH-complexes. LH4 complex was eluted with 20 mM Tris–HCl buffer solution. After washing the complex with 20 mM sodium phosphate buffer solution until complete loss of eluate colour, LL LH2 complex was eluted with 20 mM sodium phosphate buffer supplied with 0.25% Triton X-100. Then the eluates of complexes with HA were additionally purified by chromatography on DEAE-TOYOPEARL 650 S column (*Bol'shakov et al., 2016*). For this step, linear gradient ranging from 0.1 to 0.3 M NaCl was used; LL LH2 complex was eluted with 0.14 M NaCl suppplied with 0.02% DDM, and LH4 complex was eluted with 0.17 M NaCl suppplied with 0.02% DDM.

Pigment composition of LH complexes was analyzed by HPLC method (*Bol'shakov et al., 2016*). HPLC appliance (Shimadzu, Kyoto, Japan) consisted of (1) LC-10ADVP pump with FCV-10ALVP module, (2) diode matrix-based detector SPD-M20A, and (3) thermostat CTO-20 AC. Carotenoids were separated on reverse-phase column 4.6 × 250 mm (Agilent Zorbax SB-C18; Agilent Technologies, Santa-Clara, CA, USA) at 22 °C. The carotenoids were identified by retention time and absorption spectra. Quantitative determination of each carotenoid was carried out by comparing peak area from 360 to 800 nm with the sum of all the carotenoid peaks taken as 100%, and calculating the result in LC-solution program (Shimadzu, Kyoto, Japan) using molar extinction coefficients (*Britton, 1996*).

Absorption spectra were recorded on spectrophotometers Cary 50 (Agilent Technologies, USA) and Hitachi 557 (Hitachi, Tokyo, Japan). Emission and excitation spectra of fluorescence were recorded on spectrofluorimeter Cary Eclipse (Agilent Technologies, USA). Energy transfer efficiency from Car to BChl *a* was determined at the maximum absorption of carotenoids for each complex according to the method described in the work (*Niedzwiedzki, Kobayashi & Blankenship, 2011*) by comparing the fluorescence excitation spectra with absorption spectra transformed into 1-T spectra, where T is transmittance coefficient. The spectra were normalized by the peak of BChl *a* Qx-transition at 590 nm. All the spectral measurements were carried out at room temperature.

Proteomic analysis of LL LH2 complex was carried out using chromatographic HPLC-system Ultimate 3000 RSLCnano (Thermo Scientific, Waltham, MA, USA) coupled with mass spectrometer Q-Exactive HF (Thermo Scientific, Waltham, MA, USA). The accuracy of mass measurements comprised 70 ppm. LL LH2 peptides separated by one-dimensional SDS-PAGE by *Laemmli (1970)*, in 15% acrylamide gel with 0.3% bis-acrylamide and 0.1% SDS, were used as samples for mass-spectrometry. Bands with molecular weights from 10 kDa to the front line were excised and subjected to trypsinolysis. LL LH2 complexes in 20 mM Tris–HCl buffer obtained by aforementioned chromatographic methods were also studied by mass spectrometry.

All the mass spectrometry measurements were carried out using equipment of the "Human Proteome" Core Facility Centre (Institute of Biomedical Chemistry).

Models of LL LH2 were built by molecular replacement method in YASARA Structure (*Krieger et al., 2012*). For this purpose, alignments of alpha-subunits of LH2 complexes from *Rps. palustris* (Uniprot ID: P35101) and *Rbl. acidophilus* (sequences from 3D structures, PDB IDs: 1NKZ, 2FKW (*Papiz et al., 2003*; *Cherezov et al., 2006*)) were built in ClustalOmega (*Sievers et al., 2011*). Then the residues in the PDB files were substituted with the corresponding residues of alpha subunit of LH4 complex from *Rps. palustris*. The obtained PDB-file served as a source for all the following manipulations. Then, the residues of beta-subunits were substituted with the corresponding residues of beta subunits LH2A-beta ($\beta$a), LH2B-beta ($\beta$b), LH3-beta ($\beta$c), LH4-beta ($\beta$d) (Uniprot IDs: P35106, P35107, P35108, P35109). The bacteriochlorophyll ligands were conserved as they are in the initial files (complexes from *Rbl. acidophilus*), while rhodopin glucoside was substituted with rhodopin according to the experimental data on carotenoid composition.

## RESULTS

LL LH2 complex was successfully isolated in our work due to the used technique of primary isolation of LH4 and LL LH2 complexes on hydroxyl apatite (HA) column. The earlier experiments using the ion-exchange sorbents (*Tadros & Waterkamp, 1989*; *Hartigan et al., 2002*; *Taniguchi et al., 2014*) did not include sorption chromatography based on different hydrogen bonding ability. HA is a mixed-type sorbent. Its phosphate groups play a role of cation exchanger, whereas hydroxyl groups retain the separated compounds *via* formation of hydrogen bonds (*Misra, 1988*).

Spectral characteristics of the isolated LL LH2 and LH4 complexes were studied by optical spectrophotometry. The most significant differences in their absorption spectra were observed in the bands of weakly interacting $\beta$-bound monomeric BChl *a* molecules. The corresponding absorption maximum at about 800 nm in the LH4 complex is the main one. It is very narrow, which is typical for this type of complexes (*Tharia et al., 1999*; *Hartigan et al., 2002*; *Brotosudarmo et al., 2011*; *Southall et al., 2018*) (Fig. 1). In LL LH2 complex, this maximum had a split band with maxima at 797 and about 800 nm (Fig. 1 (*Evans, Hawthornthwaite & Cogdell, 1990*; *Gall et al., 1999*)). This complex also differed from LH4 complex by prevalence of absorption maximum at 850 nm, so its absorption spectrum was more in line with the LH2 complex (*Evans, Hawthornthwaite & Cogdell, 1990*; *Gall et al., 1999*; *Cogdell, Gall & Köhler, 2006*). Previously, it was shown (*Kotecha, Georgiou & Papiz, 2013*) that the ratio of the B850/B800 absorption bands in HL complexes can vary depending on illumination and *Rps. palustris* strain. However, the most significant characteristic determining the type of complex is its protein composition.

Differences in absorption spectra were also observed in positions of maxima around 370 nm (Soret-band) and 590 nm reflecting the contribution of three BChl *a* molecules of both B800 and B850 ring-shaped aggregates. Based on the analysis of the absorption spectra, it could be assumed that LL LH2 has a heterogeneous peptide composition. It could be similar to the LH4 complex, but not identical to it.

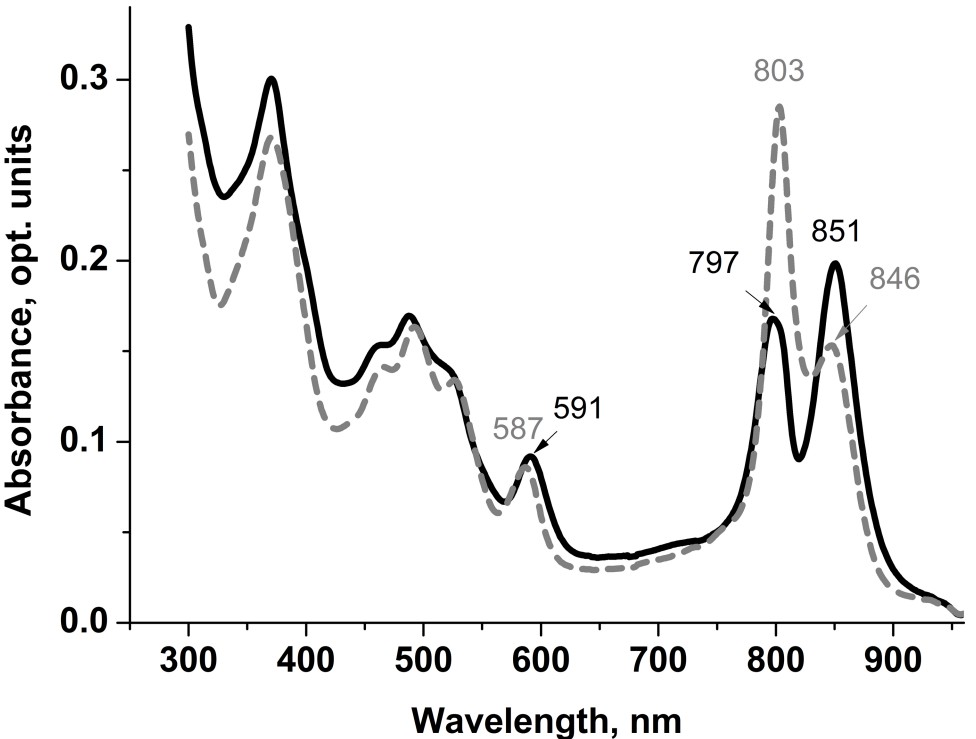

**Figure 1** **Absorption spectra of LL LH2 and LH4 from *Rps. palustris*.** LL LH2 absorption spectrum is shown in solid line, LH4 complex absorption spectrum is shown in dashed line.

Analysis of peptide samples obtained during SDS-electrophoresis of LL LH2 complex (see Materials and Methods) by HPLC-MS/MS spectrometry revealed $\beta$a sp|P35106| and $\beta$b sp|P35107| subunits of B-800-850 with real molecular weight of 5.3 and 5.7 kDa (Fig. 2, bands 1 and 2, respectively). Search of proteins corresponding to other peptide mass peaks led to reliable identification (18 peptide fragments, sequence coverage 83%) of uncharacterized protein of *Rps. palustris* (UniProt ID: Q6N9P5, molecular weight 10.8 kDa) (Fig. 2, band 3).

Three more polypeptides, $\alpha$d (P35104), $\beta$d (P35109) and $\beta$c (P35108), were detected in LL LH2 complex by tandem LC-mass-spectrometry. Amino acid sequences of all peptides found in complex LL LH2 are presented below (Table 1).

Thus, peptide composition in LL LH2 complex and LH4 complex of *Rps. palustris* strain 2.1.6 ($\alpha\beta_a$, $\alpha\beta_d$ and $\beta_b$) known from the literature (*Brotosudarmo et al., 2011*) were similar in "excessive" variability of $\beta$-subunits, but LL LH2 differed from LH4 complex by the absence of $\alpha_a$ subunit and presence of additional $\beta_c$ subunit.

Currently, there is no definite data on presence of $\beta$cd subunits in all the types of LH-complexes. These subunits were identified even in HL LH2 complex of *Rps. palustris*, but the question whether they are an impurity or a component of the complex is not discussed in the papers (*Zurdo, Fernandez-Cabrera & Ramirez, 1993*; *Tharia et al., 1999*).
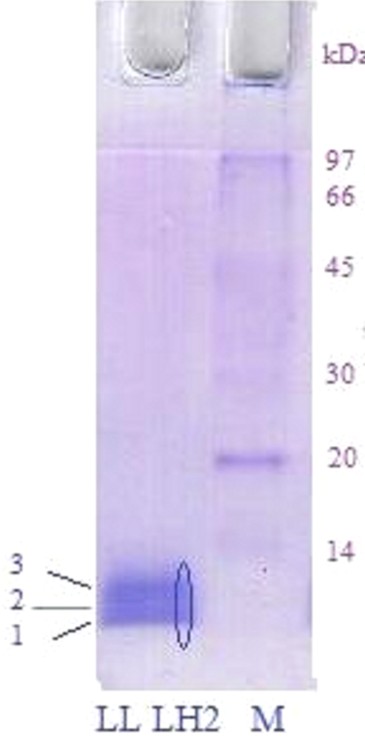

**Figure 2** **SDS electrophoresis of LL LH2 complex.** Left track—LL LH2; Right track—molecular weight markers. The oval indicates the excision of peptides along the gel for their study by HPLC MS/MS.

Current data on polypeptide composition of HL and LL complexes from *Rps. palustris* are summarized in Table 2 (*Brotosudarmo et al., 2011*) with our data added.

Differences in polypeptide composition could explain different retention degree of LL LH2 and LH4 complexes on hydroxyl apatite. Different C-terminal regions of alpha-subunits could be the determinants of different sorption activity (Fig. 3).

The difference in C-terminal sequence is presence of IESSIKAV motif in $\alpha$d in the position where $\alpha$a subunit has IESSVNVG motif. The most differing amino acids are lysine K57 (positively charged amino acid, side chain pKa = 10.53) and asparagine N57 (neutral polar amino acid with amide side chain). Besides, side chains of other residues could shield potential sorbent binding sites (side chain of S, N or K; carboxyl group of the C-terminus) differently. It should be noted, that, taken together, amino acids I, A, V in $\alpha$d are more hydrophobic, than V, V, G in $\alpha$a. All this could lead to the differences in elution ability shown upon isolation of LL LH2 and LH4: LH4-complex was eluted by 20 mM Tris–HCl, whereas LL LH2 was eluted by 20 mM sodium phosphate with 0.25% Triton-X100. In this case, phosphate ions apparently interacted with lysine residue better than chloride ions, tearing it off the phosphate groups of hydroxyl apatite, while the added detergent presumably prevented the aggregation of hydrophobic residues of the C-termini of alpha-subunits between each other and with other hydrophobic components retained at the start (at the top of the column).

Serdyuk et al. (2023), *PeerJ*, DOI 10.7717/peerj.14769

**Table 1  Proteins detected in LL LH2 complex of *Rps. palustris* by mass-spectrometry.**

| Uniprot ID | Uniprot description | Raw sequence | Brief name |
|---|---|---|---|
| P35104 | LHA4_RHOPA Light-harvesting protein B-800-850 alpha chain D | MNQGRIWTVVKPTVGLPLLLGSVAIMVFLVHFAVLTHTTWVAKFMNGKAAAIESSIKAV | $\alpha$**d** |
| P35106 | LHB1_RHOPA Light-harvesting protein B-800-850 beta chain A | MADKTLTGLTVEESEELHKHVIDGTRIFGAIAIVAHFLAYVYSPWLH | $\beta$**a** |
| P35107 | LHB1_RHOPA Light-harvesting protein B-800-850 beta chain A | MADKTLTGLTVEESEELHKHVIDGTRIFGAIAIVAHFLAYVYSPWLH | $\beta$**b** |
| P35108 | LHB3_RHOPA Light-harvesting protein B-800-850 beta chain C | MVDDSKKVWPTGLTIAESEEIHKHVIDGARIFVAIAIVAHFLAYVYSPWLH | $\beta$**c** |
| P35109 | LHB4_RHOPA Light-harvesting protein B-800-850 beta chainD | MVDDPNKVWPTGLTIAESEELHKHVIDGSRIFVAIAIVAHFLAYVYSPWLH | $\beta$**d** |
| Q6N9P5 | Uncharacterized protein | MSEEYKGHSGHPLILKQEGEYKGYSGEPLILKQEGEYKGYSGTPL ILEQKGEYQSFSGTPLILKQEGEYRGFSGAPLILKQDGEYKSFSGYPLLLNI | $\gamma$ (**?**) |
**Table 2  Data on polypeptide composition of LH2, LH4 and LL LH2 complexes of *Rps. palustris*.**

| Data source | LH2 | LH4 | LL LH2 |
|---|---|---|---|
| *Tadros & Waterkamp (1989)* | PucABa, PucABb, PucABc, PucABd | – | – |
| *Tharia et al. (1999)* | PucABa, PucABb, PucABd | PucABa and PucABd | – |
| *Brotosudarmo et al. (2011)* | PucABa and PucABb | PucABa, PucABd, PucBb | – |
| Present study | – | – | Puc ABd, PucBabc |

```
P35101   MNQARIWTVVKPTVGLPLLLGSVTVIAILVHFAVLSHTTWFSKYWNGKAAAIESSVNVG-   59
P35102   MNQGRIWTVVNPGVGLPLLLGSVTVIAILVHYAVLSNTTWFPKYWNGATVAAPAAAPAPA   60
P35103   MNQGRIWTVVSPTVGLPLLLGSVAAIAFAVHFAVLENTSWVAAFMNGKSVAAAPAPAAPA   60
P35104   MNQGRIWTVVKPTVGLPLLLGSVAIMVFLVHFAVLTHTTWVAKFMNGKAAAIESSIKAV-   59
         ***.******.* **********: :.: **:*** :*:*.  : ** :.*   : .

P35101   ------   59
P35102   APAAKK   66
P35103   APAKK-   65
P35104   ------   59
```

**Figure 3  Multiple alignment of alpha-subunits of *Rps. palustris* light-harvesting complexes.** The sequence differences in the C-terminal motif are underlined. The proteins are designated by their Uniprot IDs: P35101, $\alpha$a; P35102, $\alpha$b; P35103, $\alpha$c; P35104, $\alpha$d. The colours used are from the standard ClustalOmega colour code: red, hydrophobic (A, V, I, L, M. F, W, P); green, polar neutral (G, S, T, Q, N, Y, H); magenta, positively charged (R, K); blue, negatively charged (D, E).

Another important characteristic of LH complexes is composition and content of carotenoids present. The composition of carotenoids in the LH4 and LL LH2 complexes isolated from *Rps. palustris* grown under very low light conditions (10 lux) was studied. A distinctive feature of LL LH2 was the different level of carotenoid biosynthesis from the LH4 complex (Table 3). As can be seen from Table 3, significant differences are observed in the ratio of all carotenoids in the two LL complexes, with the exception of anhydrorhodovibrin, which is present in them in an equal percentage.

In the LH4 complex, the main carotenoids were lycopene and rhodopin and in a typical ratio, as described in the literature (*Gall et al., 2005*; *Brotosudarmo et al., 2015*; *Hashimoto, Uragami & Cogdell, 2016*; *Muzziotti et al., 2017*). The LL LH2 complex contained almost an order of magnitude less lycopene, and 1.5 times more rhodopin. Both LL complexes differed from the classical LH4 (*Brotosudarmo et al., 2009*; *Brotosudarmo et al., 2015*) by rather high content of 3,4-didehydrorhodopin, the LL LH2 complex was distinguished by a relatively high level of spirilloxanthin, which was absent in the LH4 complex. The detailed data of HPLC analysis are submitted as File S1.

Apparently, selective distribution of carotenoids in LH4 and LL LH2 complexes affected the energy transfer efficiency from Car to BChl *a*. Light-harvesting complexes are excited at wavelength corresponding to carotenoid absorbance, and energy transfer to bacteriochlorophyll is detected by monitoring of its fluorescence in red or near-infrared spectral area. The analysis of excitation spectra of LH complex fluorescence in *Rps. palustris* showed that the efficiency of energy transfer from Car to BChl in LL LH2 is significantly lower than in LH4 complex (Fig. 4).

![PeerJ]

**Table 3** Carotenoid composition of LL LH2 and LH4 complexes from *Rps. palustris* cells grown at low light intensity.

| Carotenoid | Retention time, min | Content, mol % | |
|---|---|---|---|
| | | LL LH2 | LH4 |
| Demethyl-spirilloxanthin | 14.4 | 0.6 | 1.1 |
| Rhodovibrin | 15.2 | 0.3 | 2.2 |
| 3,4-didehydrorhodopin | 19.1 | 17.4 | 32.5 |
| Rhodopin | 20.1 | 65.8 | 43.4 |
| Spirilloxanthin | 20.6 | 9.6 | – |
| Anhydrorhodovibrin | 26.1 | 4.6 | 4.6 |
| Lycopene | 31.4 | 1.7 | 16.2 |

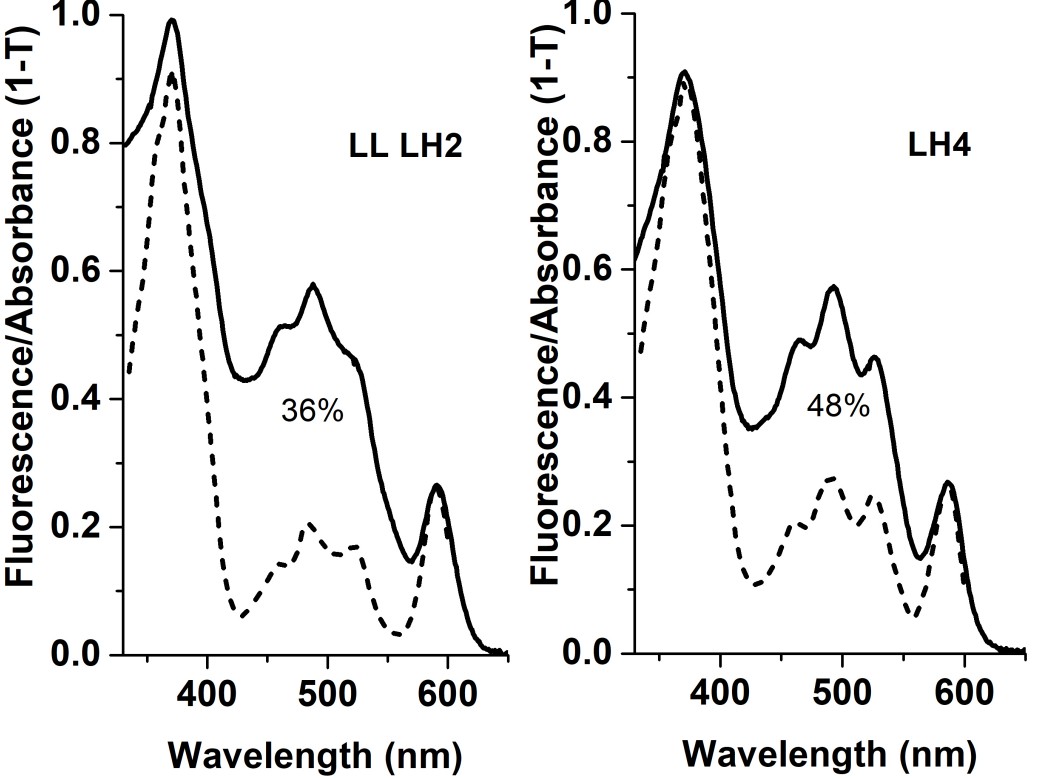

**Figure 4** Fluorescence excitation spectra (the dotted lines) and the absorption spectra (in the form of 1-T, solid line) of the LH2 complexes from *Rps. palustris.* The spectra were normalized by BChl *a* Qx-transition peak at 590 nm. The emission wavelengths for the fluorescence excitation spectra of LL LH2 and LH4 were 864 and 861 nm, respectively. The percentages written in the figure correspond to energy transfer efficiency from carotenoids to BChl in these complexes.

This process could be affected by such factors as carotenoid orientation in relation to the adjacent BChl molecules, structure of polypeptides composing the LH complexes and structure of carotenoids themselves (*Papagiannakis et al., 2003*; *Cong et al., 2008*; *Frank & Polívka, 2009*; *Chi et al., 2015*; *Ashikhmin et al., 2017*).

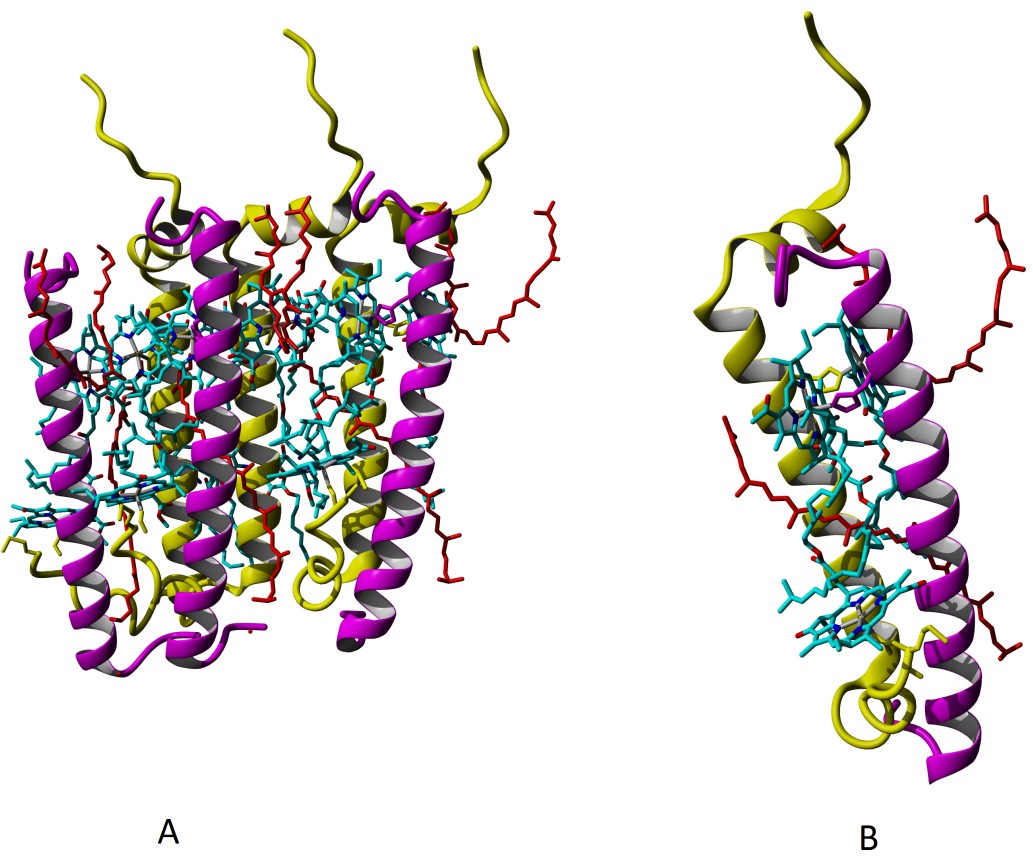

A    B

**Figure 5** **Models of LL LH2 complex based on 1NKZ template (LH2 complex of *Rbl. acidophilus*).**
Rhodopin is colored red; bacteriochlorophyll is colored *via* conventional element scheme—carbon, cyan;
nitrogen, blue; oxygen, red; magnesium, yellow; alpha subunits are yellow, beta subunits are magenta. (A)
Hexameric view; (B) close view of a single alpha-beta dimer with two rhodopin molecules and three bacte-
riochlorophyll molecules.

It is a common belief that the major carotenoid stabilizing the structure of peripheral
LH complexes in *Rbl. acidophilus* and *Rps. palustris* is rhodopin glucoside. Based on the
experimental data for LH3 and LH2-complexes of *Rbl. acidophilus*, structural modeling
of LL LH2 complex with this carotenoid was carried out, but the glycosyl residues were
deleted (Fig. 5), taking into account our data on presence of rhodopin instead of rhodopin
glucoside in LL LH2 complex (see Table 2).

To study the mutual relation between composition of peptides and chromophores and
LL LH2 functioning, we performed multiple alignment of $\beta$-subunit sequences detected
in LL LH2 complex, and demonstrated interaction of various amino acid residues with
Car and BChl *a* on three-dimensional models. It is known that the specific subunit of LH4
complex is $\beta$d, and the differences between primary sequences of $\beta$-subunits are slight
(Fig. 6). To reveal the impact of certain amino acid residues on the differences in spectral
properties and energy transfer efficiency, their positions in relation to chromophores were
analyzed.

```
P35106    ----MADKTLTGLTVEESEELHKHVIDGTRIFGAIAIVAHFLAYVYSPWLH    47
P35107    MADDPNKVWPTGLTIAESEELHKHVIDGTRIFGAIAIVAHFLAYVYSPWLH    51
P35108    MVDDSKKVWPTGLTIAESEEIHKHVIDGARIFVAIAIVAHFLAYVYSPWLH    51
P35109    MVDDPNKVWPTGLTIAESEELHKHVIDGSRIFVAIAIVAHFLAYVYSPWLH    51
          .    ****: ****:*******:*** *****************
```

**Figure 6** **Multiple alignment of beta-subunits of peripheral LH-complexes in *Rps. palustris*.** The proteins are designated according to their UniProt IDs: P35106, βa; P35107, βb; P35108, βc; P35109, βd. Positions of variable amino acids of beta-subunits are underlined. The colours used are from the standard ClustalOmega colour code: red, hydrophobic (A, V, I, L, M. F, W, P); green, polar neutral (G, S, T, Q, N, Y, H); magenta, positively charged (R, K); blue, negatively charged (D, E).

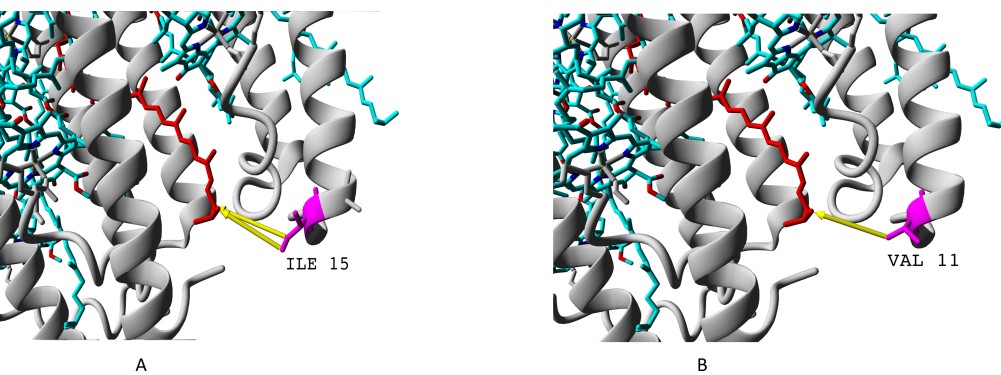

**Figure 7** **(A–B) Differences of valine residue V11 and isoleucine residue I15 in interaction with rhodopin.** Rhodopin is coloured red. Bacteriochlorophyll is coloured *via* standard element scheme (cyan, carbon; blue, nitrogen; red, oxygen; yellow, magnesium). Valine resiude V11 (specific for βa subunit) is coloured magenta; isoleucine I15 (specific for βbcd subunits) is coloured orange. Black arrow shows hydrophobic contact common for valine and isoleucine (distance 6.477 Å); green arrow depicts a contact unique for isoleucine residue (distance 6.546 Å).

We showed presence of all the four variants of beta-subunits in LL LH2 complexes, which is different from LH4 complex lacking βc subunit according to data of Tharia and colleagues 1999; the fraction of α βa peptides in LH4 complex is significantly lower, than the fraction of α βd peptides. That is why we examined differences between βa and βd and between βc and βd when comparing LL LH2 and LH4 complexes. Figs. 7–10 show superpositions of LH4 complex with LL LH2 complex including βa and βc subunits. The distances between chromophores and corresponding amino acid residues were of special interest. According to Onofrio, 2014, hydrophobic contacts in range from 3.8 to 9.5 Å between aliphatic carbon atoms are the most widespread in protein structures (*Onofrio et al., 2014*). The differences between valine V11 and isoleucine I15 were observed right in this range (Fig. 7).

As it can be seen, the presence of valine residue instead of isoleucine could weaken interaction of the βa subunit with rhodopin due to the absence of hydrophobic contact specific for isoleucine residue (*via* distal atom of sec-butyl side chain).

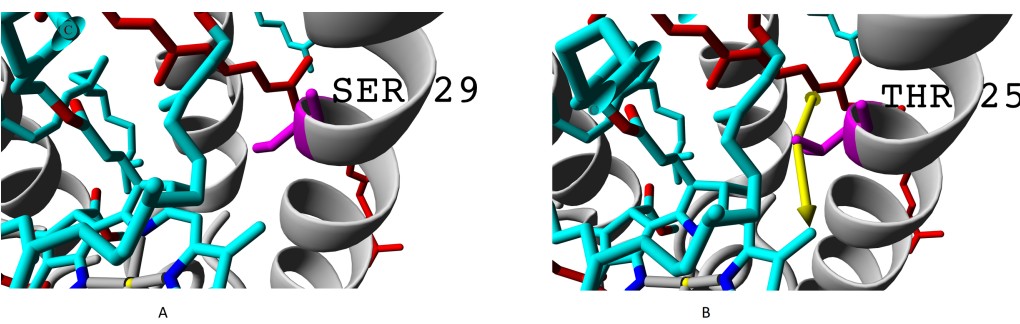

**Figure 8** **(A) Serine and (B) threonine residues and the difference in their interaction with chromophores.** Alpha subunits are coloured yellow, beta subunits are coloured magenta. Rhodopin is coloured red, bacteriochlorophyll B850 is coloured according to the standard element colouring scheme (carbon, cyan; nitrogen, blue; oxygen, red; magnesium, yellow). Threonine residue T25 specific for $\beta$ab subunits is coloured orange; serine residue S29 specific for $\beta$d subunit is coloured magenta Green arrows show the contacts of threonine methyl group with bacteriochlorophyll (distance 3.955 Å) and rhodopin (3.866 Å).

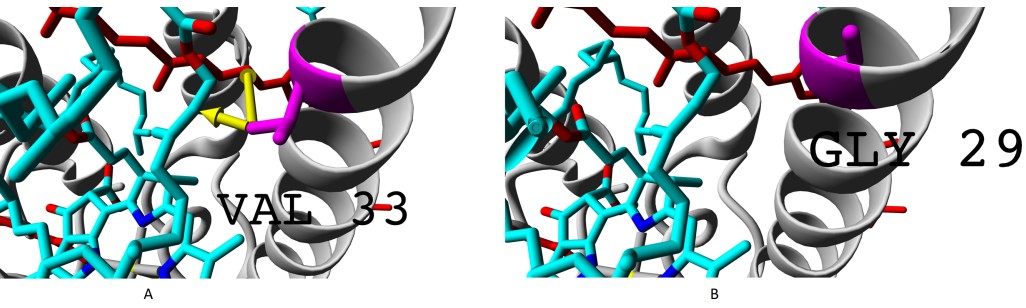

**Figure 9** **(A–B) Valine residue V33 of LH4 complex and its interaction with chromophores (rhodopin, coloured red, and bacteriochlorophyll, coloured *via* standard element coloring scheme).** Green arrow shows interaction with rhodopin (distance 6.958 Å); black arrow shows possible van der Waals repulsion from bacteriochlorophyll (distance 2.370 Å).

Substitution of alanine A16 in LH4 to glutamate E12 in LL LH2 has most probably no effect on the spectral properties and energy transfer, because these residues do not interact with chromophores.

Even more important difference than presence of valine instead of isoleucine is presence of threonine or alanine on the position of serine. Hydroxyl group of serine residue does not participate in formation of hydrogen bonds (neither with alpha-subunit nor with chromophores). Presence of additional methyl group in threonine makes this amino acid position more involved into interaction with both bacteriochlorophyll B850 and rhodopin (Fig. 8).

Presence of glycine residue G29 in place of valine V33 could also play a significant role. Valine residue interacts with both rhodopin and bacteriochlorophyll. Meanwhile, in the built model, interaction with bacteriochlorophyll takes place at extremely short distance, which could lead to van der Waals repulsion (Fig. 9).

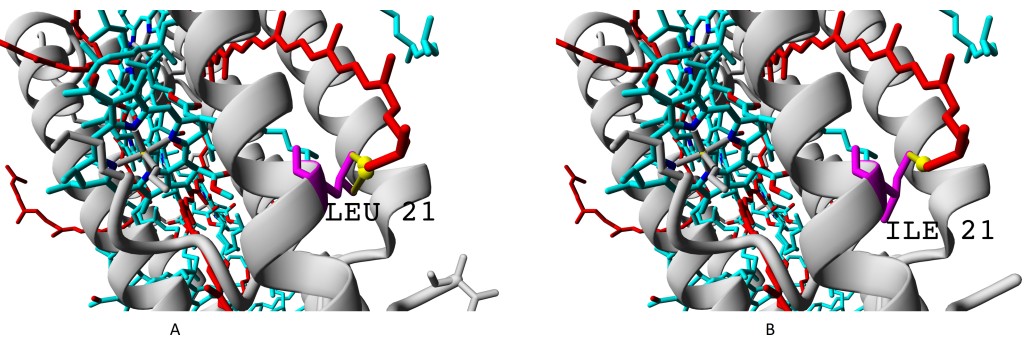

**Figure 10** **(A–B) Isoleucine residue I21 (specific for βc subunit) and leucine residue L17/L21 (specific for βabd subunits), differences in interactions with chromophores (rhodopin, colored red).** Bacteriochlorophyll is colored according to standard element scheme. Black arrow shows a contact common for isoleucine and leucine residues with rhodopin (distance 5.155 Å); green arrow shows a contact unique for leucine residue (distance 6.599 Å).

The van der Waals repulsion of bacteriochlorophyll could lead to somewhat different geometry of chromophores in the complexes containing subunits βc and βd, and this, in turn, could lead to lower efficiency of energy transfer between rhodopin and bacteriochlorophyll. The mechanism of this phenomenon requires more thorough examination, but now we could suppose a conformational transition of bacteriochlorophyll causing the change of slope angle of tetrapyrrol plane in relation to rhodopin axis.

The residue unique for beta-subunit βc is shown in Fig. 10.

Isoleucine could interact with rhodopin weaker than leucine, which could be reflected in spectral properties and/or energy transfer efficiency.

Thus, the mapped differences in amino acid sequences could explain the different spectral properties and energy transfer efficiency.

## DISCUSSION

Light, the main energy source in purple bacteria, is absorbed primarily by light-harvesting LH2 complexes and then transferred to LH1-RC complex with further conversion of energy into chemical bond energy in chromatophores (*Fejes et al., 2003*; *Scheuring et al., 2006*). Variability of peptide and carotenoid composition in light-harvesting complexes makes them unique systems of adaptation of phototrophic purple bacteria to different illumination conditions.

The results of the current work provide more reliable evidence for the existence of the LL LH2 complex along with the LH4 complex, than the earlier study (*Serdyuk, Smolygina & Ashikhmin, 2020*). Data on the composition of proteins and carotenoids of the LL LH2 complex were obtained, which distinguish it from all the other known LH complexes (*Zhou, Zhang & Zhang, 2014*; *Lopez-Romero et al., 2020*). Three-dimensional modeling showed which amino acid residues of all βabcd subunits found in this complex can interact with Car rhodopin and BChl *a*; the difference between LH 4 and LL LH2 complexes in the efficiency of energy transfer from Car to BChl *a* was shown.

According to the literature, single-molecule spectroscopy of LH4 complexes in the narrow LL illumination range from 200 lx to 5.5 lx showed gradual increase of 800/850 absorption ratio and constriction of Qy band of B850. This was observed to 10 lx value, while further decrease of illumination led to the increase of Qy 800/850 ratio and broadening of the peak at 850 nm. 10 lx was a margin of characteristic spectral changes upon the decrease of illumination (*Brotosudarmo et al., 2009*).

Interestingly, in our earlier work (*Serdyuk, Smolygina & Ashikhmin, 2020*), the LH4 and LL LH2 complexes from *Rps. palustris* grown under 50 lux illumination had almost the same ratio of carotenoids. In the present study, these complexes differed significantly in this parameter. The inconsistency of the data could be explained by the insufficiently careful separation of the complexes into HA in the first study, but this assumption is refuted by comparable data on Car for LH4 complexes in the earlier and present work. Most likely, these differences in Car are associated with the use of different illumination in two studies (50 lx and 10 lx) and, to some extent, with additional purification of LL LH2 in this study by HPLC. The composition of the proteins from LL LH2 complex was not investigated in our earlier work (*Serdyuk, Smolygina & Ashikhmin, 2020*). The successful isolation of LL LH2 complex was possible due to the used technique of primary isolation of LH4 and LL LH2 complexes on HA column. The validity of this statement was confirmed by the analysis of the amino acid sequences of the $\alpha$-subunits of both LL-complexes, which showed possible determinants of different retention force for LL LH2 and LH4 on hydroxylapatite as different C-terminal motifs in the $\alpha$d-subunit. Multiple alignment of the alpha-subunits (see 'Results', Fig. 2) showed that the principal difference between $\alpha$a and $\alpha$d subunits is located at the C-termini. Moreover, it should be taken into account that the isolated complexes are membrane-bound. Hence, the major part of the polypeptide chain is shielded from the interaction with sorbents by lipids. That is why the molecular reason for different retention force of LL LH2 and LH4 could be the difference in the C-termini of $\alpha$-subunits. In the previous studies (*Tadros & Waterkamp, 1989*; *Hartigan et al., 2002*; *Taniguchi et al., 2014*) application of ion exchange sorbents did not include sorption chromatography based on different abilities to form hydrogen bonds. HA is a mixed-type sorbent, its phosphate groups play a role of cation exchange sorbent (*Itoh, Yoshimoto & Yamamoto, 2018*), whereas hydroxyl groups retain the separated proteins *via* hydrogen bonding (*Misra, 1988*).

Differences in the polypeptide composition of LL LH2 and other complexes are the main features of its individuality and evidence of the biosynthesis of a new type of peripheral light-harvesting complex under LL conditions. Mass spectrometry of proteins of the LL LH2 complex showed their difference from the composition of LH4 complex ($\alpha\beta$a, $\alpha\beta$d, and only $\beta$b) demonstrated earlier (*Brotosudarmo et al., 2011*). In this work, an even greater excess of the number of $\beta$-subunits in relation to alpha subunits was shown, namely the presence of $\beta$c, and the absence of $\alpha$a-subunits. The observed absence of $\alpha$b-subunits in LH4 complex in the work of *Brotosudarmo et al. (2011)* and absence of $\alpha$ab subunits in LL LH2 in our work may be explained by the loss of their functional importance at LL illumination conditions. The ability to shift from B800–850 to B800 could increase the efficiency of RC functioning, as shown in LL-adapted complex B800–820 (*Deinum et*

*al., 1991*), where B800–820 was an effective barrier against reverse energy transfer from LH1/RC. Such energy transfer from LH1/RC to HL LH2 complex occurs under high illumination to prevent oxidation of RC (*Freiberg et al., 1996*).

Higher variability of $\beta$-subunits compared to $\alpha$-subunits was also observed in another work (*Tharia et al., 1999*), but the authors could not detect $\beta c$ subunit in LH4 complex neither in HL nor in LL samples. They supposed that its synthesis is mainly caused by other stimuli like temperature, and it was not detected in their cultivation conditions (t = 22 °C). Other authors detected $\beta c$-subunit in HL LH2 from *Rps. palustris*, but they did not demonstrate reliable proofs of its presence (*Tadros & Waterkamp, 1989*) (See 'Results', Table 2). Other researchers supposed that it is impossible to isolate LH3 and LH4 complexes from each other due to high identity degree of primary sequence of their proteins (*Brotosudarmo et al., 2011*). The possibility of existence of different LH4 complexes with various $\beta$-subunits in *Rps. palustris* might find indirect proof in the work on single-molecule fluorescence of these complexes (*Ilioaia et al., 2018*), where the presence of at least four one-ring "subpopulations" with high number of conformational subvariants and configurations created by an unknown chaperone-like process was shown. The authors suggest that addition of different types of PucB apoprotein in different ratios into the B850 "rings" is a very effective strategy for increasing the light harvesting ability under stressful LL illumination conditions (*Ilioaia et al., 2018*).

Absorption spectra of LH2 complexes in the near infrared (NIR) region depend on pigment-protein interactions, which determine the energy levels of pigments, and on pigment-pigment interactions, which determine exciton interactions (*Taniguchi et al., 2014*). Taking into account the differences in the carotenoid region of the absorption spectra of the two LL complexes, as well as the differences in the protein composition, we conducted a study of the carotenoid composition. In the present study, the use of additional purification of LL LH2 and LH4 complexes on a DEAE-TOYOPEARL 650 S HPLC column and a change in the light regime of cell growth(10 lx) revealed significant differences between their carotenoid ratios and levels of energy transfer efficiency, distinguishing them from the same complexes obtained earlier (50 lx) (*Serdyuk, Smolygina & Ashikhmin, 2020*). A distinctive feature of the LL LH2 complex is very high content of rhodopin (66%), comparable to that of the HL LH2 complex (75%) (*Brotosudarmo et al., 2015*) and high content of 3,4-didehydrorhodopin in both complexes, which was not previously noted in the literature, and which is probably a characteristic feature of carotenoid biosynthesis in the bacterium *Rps. palustris* 1e5. The presence of such a high amount of rhodopin, its derivatives 3,4-didehydrorhodopin and spirilloxanthin, in the amount of 83% of all other carotenoids of this complex, determines very low levelof lycopene (1.7%), which is an intermediate product in the synthesis of these xanthophyllic carotenoids.

However, we did not observe the same differences in carotenoids and energy transfer efficiency from carotenoids to BChl a in the two LL complexes in our earlier work. The discrepancy between the results in our two works can be explained by different light conditions (50 and 10 lx) for growing the cultures from which the complexes were obtained, or by the absence of additional purification step of the complexes in the previous work.

Thus, rhodopin and 3,4-didehydrorhodopin are the main carotenoids in LH4 and LL LH2 *Rps. palustris*, as shown in this work and in our previous one (*Serdyuk, Smolygina & Ashikhmin, 2020*). Since lycopene is an intermediate in the biosynthesis of the above two carotenoids, its content in the complexes depends on the need of cells for the two aforementioned carotenoids, as was shown by us and other researchers (*Brotosudarmo et al., 2011*). Apparently, lycopene plays a role of source pool, and it is used for the synthesis of other carotenoids when needed. Rhodopin and 3,4-didehydrorhodopin perform well-known functions specific for carotenoids in LH complexes. The crystal structure of the LH2 complex shows that carotenoids and B800 molecules interact closely (*Papiz et al., 2003*; *Gall et al., 2005*). Taking into account the excessive variability of $\beta$-subunits compared to $\alpha$-subunits in LH4 complex (*Brotosudarmo et al., 2011*) and in LL LH2 complex shown in our work, we can suppose that this is the reason for binding of different amounts and relations of carotenoids in these complexes.

In the present study, LL LH2 was found to be isolated together with uncharacterized protein Q6N9P5. The functional role of this protein and its homologs has been unknown and requires further examination. During the preparation of this manuscript, some interesting data on cryo-electron microscopy of light-harvesting complexes were submitted to Protein Data Bank database (PDB IDs: 7ZCU, 7ZDI, 7ZE8, 7ZE3). According to them, this protein forms a crown around the cytosolic side of LH-complexes, and it could participate in arrangement of bacteriochlorophyll molecules, performing a function of a scaffold aggregating six extra bacteriochlorophyll molecules around the LH2 complex. Then, an article was published in PNAS on these experimental data, where the authors proposed to designate this protein as gamma-subunit of light-harvesting complexes (*Qian et al., 2022*).

## CONCLUSIONS

It is known that different types of peripheral light-harvesting complexes known up to date are characterized by different polypeptide composition, content and composition of carotenoids, energy transfer efficiency from Car to BChl *a*. A new type of complex LL LL2, isolated by us from *Rps. palustris*, 1e5 strain, differed from other known LH complexes in all of the above physicochemical parameters.

The LL H2 complex differed from the LH4 complex by the presence of an additional $\beta$c subunit, which is commonly specific for LH3 complex, and a low level (36% *versus* 48%) of the efficiency of energy transfer from Car to BChl *a*, respectively. The main carotenoids were rhodopin and 3,4-didehydrorhodopin, in a ratio of 43/32 in LH4 and 66/17 in LL LH2. To study the mutual relation between composition of peptides and chromophores and LL LH2 functioning, we performed multiple alignment of $\beta$-subunit sequences detected in LL LH2 complex. Multiple alignment of sequences of $\beta$-subunits of the LL LH2 complex and its three-dimensional modeling showed the interaction of isoleucine I21, specific for $\beta$c, and leucine L17/L21, specific for the $\beta$abd subunit, with rhodopin. A weaker interaction of isoleucine than leucine with rhodopin may affect the spectral properties of the LL LH2 complex and the efficiency energy transfer.

High adaptability of *Rps. palustris* to various light conditions is associated not only with its multiple *puc* genes responsible for the synthesis of LH complex proteins, but also with a complex system of regulation of their biosynthesis by the same variety of bacteriophytochromes. Previously, it was shown that when shifting from HL to LL conditions in *Rps. palustris*, unlike *Rbl. acidophilus*, an LH4 complex is formed with a different composition of polypeptides than in the LH2 complex. In this work, we have shown that another LL LH2 complex is formed along with LH4 under very low illumination, with an even more complex protein structure.

## ACKNOWLEDGEMENTS

All the mass-spectrometry measurements were carried out using equipment of the "Human Proteome" Core Facility Centre (Institute of Biomedical Chemistry). The authors thank the employees of the Centre, Olga Tikhonova and Victor Zgoda, for their contribution.

### Funding

The work was funded by the Russian Ministry of Science and Higher Education (Registration Number: 1220411200039-0). The data on carotenoid composition of light-harvesting complexes (Table 3) were obtained with financial support from the grant of the President of the Russian Federation for the state support of young Russian scientists—PhD (grant number MK-1352.2021.1.4). The funders had no role in study design, data collection and analysis, decision to publish, or preparation of the manuscript.

### Grant Disclosures

The following grant information was disclosed by the authors:
Russian Ministry of Science and Higher Education: 1220411200039-0.
President of the Russian Federation for the state support of young Russian scientists—PhD: MK-1352.2021.1.4.

### Competing Interests

The authors declare there are no competing interests.

### Author Contributions

- Olga Petrovna Serdyuk conceived and designed the experiments, performed the experiments, analyzed the data, prepared figures and/or tables, authored or reviewed drafts of the article, and approved the final draft.
- Azat Vadimovich Abdullatypov conceived and designed the experiments, performed the experiments, analyzed the data, prepared figures and/or tables, authored or reviewed drafts of the article, and approved the final draft.
- Lidiya Dmitrievna Smolygina conceived and designed the experiments, performed the experiments, prepared figures and/or tables, and approved the final draft.

- Aleksandr Aleksandrovich Ashikhmin conceived and designed the experiments, performed the experiments, analyzed the data, prepared figures and/or tables, authored or reviewed drafts of the article, and approved the final draft.
- Maxim Alexandrovich Bolshakov conceived and designed the experiments, performed the experiments, prepared figures and/or tables, and approved the final draft.

## Data Availability

The raw data, list of proteins detected in electrophoresis gel eluates and list of proteins detected in the sample after elution from HA column are available in the Supplemental Files.

## Supplemental Information

Supplemental information for this article can be found online at http://dx.doi.org/10.7717/peerj.14769#supplemental-information.

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
