# Peer review of "Simultaneous functioning of different light-harvesting complexes—a strategy of adaptation of purple bacterium Rhodopseudomonas palustris to low illumination conditions"

_PeerJ, doi:10.7717/peerj.14769_

## Round 0.1 · original submission · Major Revisions

We have received three very detailed high-quality reports on your work, with plenty of suggestions for the improvement of the presentation of your data. We hope you will soon be able to resubmit an updated version of this interesting work.

·

Basic reporting

The article by Serdyuk et al presents an improved characterization of a low light-specific complex from Rhodopseudomonas palustris, LL LH2. This complex was isolated from the authors in a previous study; they now determined its peptide composition, provide a new table for its carotenoid content, and use structure modeling to provide a tentative link between the different peptides in LL LH2 and LH4 complexes and the spectral differences observed.
Altogether the results are interesting. However, the article could be greatly improved for clarity and consistency by reorganization and some rewriting, with a stated goal more consistent with the results presented and an extended discussion of the new results and the discrepancies with the ones previously published.

The English is acceptable, but the article needs heavy proofing, as well as some rephrasing in parts:
- Line 33, Rps should be fully developed, as it is its first occurrence in the main text.
- Line 64, for clarity and ease of read, please replace “These two bacteria” by species names.
- Line 113, “LH2 complex was eluted with 0.14 M NaCl” should be removed, as that complex is not mentioned anywhere in the results.
- Line 154, “We suggest that LL LH2 was successfully isolated in our work” is an odd phrasing, especially to start the results part. It could be changed by “We previously successfully isolated the LL LH2 complex”, or a similar sentence.
- Line 167, “…had a shoulder about 800 nm”: it is not really a shoulder. I suggest to use another sentence to describe it, such as “the peak was broader, with increased absorption at 800 nm”.
- Lines 174-175: absorption spectra are not enough to say that LL LH2 “has heterogenic peptide composition”. The sentence should be shortened to “Based on the analysis of absorption spectra, it could be suggested that LL LH2 and LH4 have different peptide composition”.
- Lines 187 to 209: Rather than pasting raw data in the middle of the text, the authors should include this in a table with appropriate annotation and description.
- Line 248: “the level of carotenoid biosynthesis” should be “the amount of carotenoids”, or “the carotenoid composition”
- Lines 253-259: These lines should be removed or moved to discussion section.
- Line 425: “biosynthesis” should be “binding”
- Line 433, “excessive” should be “important”
- In the legend of Figure 5, “Different amino acids” would better be “sequence differences”

Finally, many errors and typos are remaining. A non-extensive list is below:
- Line 56, “spectroscophy” should be “spectroscopy”
- Line 128, “spactrea” should read “spectra”
- Line 134, “SDS-PAAG” should be “SDS-PAGE”
- Species names should be in italics, lines 143,146-147, 151, 445
- Throughout the text, Table 2 is never mentioned as such and generally written as Table 1. This should be corrected, for instance lines 249, 280, and above the table itself.
Different fonts are also used throughout the paper, for instance lines 223, 227-228, 295, 296, 374, 378

Concerning figures, some important improvements should be done:
- For Figure 2 and Figure 5, a figure of better quality should be provided, with a properly annotated and described alignment
- For Figure 3, more information should be added to the legend: emission wavelength for the fluorescence excitation spectra and an explanation of the percentages written in the figure, at the very least.
- For Figure 4, more information should be added to the legend, such as a description of the colors – molecules correspondence. The structure should be better described, and, for clarity, the alpha and beta chains identified, possibly by different colors. The figure could also win in clarity if a single AB pair - or two - was represented, instead of, I guess, 3 identical pairs here?
- For Figure 6, the colors used in description (e.g. cyan and green) should not appear in other parts of the picture as well.
- For Figure 7, for clarity and ease of read, it would be better to use a more simple color scheme, and especially replace the standard element coloring scheme by a uniform color for the S29 residue; it is enough then to state as a note that the Ser and Thr residues are superposed. As for the CXM alpha part, it is not mentioned in any other part of the paper and should be either removed or properly addressed.
- In general for Figures 6 to 9, it would be better to add pictures from different angles, so the structure can be more easily visualized.
- A Figure should be added for the results corresponding to the peptide analysis (lines 176 to 186): the results provided as raw data should also be properly formatted and presented in the main text.

As for the raw data supplied, it has been provided, but without the least formatting or description, making very difficult to understand what they correspond to. It should be properly organized and checked for legibility: in the HPLC data for instance, the peaks results table is pasted without even fully extending one of the columns, making its title only partially legible.

Experimental design

The research is within the scope of the journal, and the techniques applied appropriate to answer the questions raised.

However, the article could be improved in clarity and consistency.
The statement on the goal of the study (lines 89-91), in particular, should be changed. It is written that the technique for the isolation of LL LH2 was improved, but it is not mentioned how in any way and no further comment on this is made on the paper. This information should be clearly identified.
In the same manner, it is stated line 91 that “its physiochemical characteristics were revised”, but no comment on the differences (discrepancies?) with the previous analysis published by the authors in 2020 is made in the rest of the article.
To the contrary, the authors should also mention here their modeling work to link sequence differences in beta chains to spectral differences, which is not mentioned at all although it is a large part of the article.

As for another minor modification to be brought to the methods section:
- Line 134, “by Laemmli” should be “using Laemmli protocol”. The authors should then precise at least which acrylamide concentration was used (%T and %C).

Validity of the findings

The results are supported by the data, but number of them need a better description and discussion.
- As a general rule, the findings for LL LH2 and LH4 are often compared to LH2 through the paper. It would be thus interesting if this complex was added to the characterization, or if the authors could more systematically point to references in which the data can be found. An example is line 163, “its absorption spectrum corresponded to LH2 complex but with higher contribution of 800 nm absorption band”.
- Lines 235 on: the carotenoid composition presented here is significantly different to the one already published by the authors (Serdyuk et al, 2020). The discrepancy should be properly addressed and discussed.
- The presence of rhodopin rather than rhodopin glucoside was already demonstrated in the previous article published by the authors in 2020. This fact should be clarified here. Also for this reason, it is unnecessary and confusing to present a model structure with rhodopin glucoside in Figure 4, panel A. This panel should be removed, unless the model is further discussed: could the different peptide composition explain the difference in carotenoid content?
- The structural and biological significance of the different carotenoid composition should also be further discussed. Would the other carotenoids replace rhodopin in the structure? Is there a possibility that some sequence differences, rather than simply affect rhodopin coordination, promote its replacement by another molecule, providing another explanation for the spectral differences?

·

Basic reporting

(1) There is no clear definition structure between result and discussion. The part of results should present the result of analysis in an objective manner and should not heavily discuss the result, example: line 154-160, line 211-215, line 235-247, line 253-259, line 271-274.
(2) Table 1 and Table 2 have the same caption as “Table 1”.
(3) Table 1 Data on polypeptide… is suitable for discussion because it has compared the result with previous published data.

Experimental design

(1) The goal of research (line 89) to reveal data on protein composition, however this is only one of the goals, and the conclusion clearly indicates there are more than one goal. Therefore the research question was not well defined. The sentence need to be revised.
(2) Line 97-100 method for cell growth, please indicate the detail information to define stationary phase and how to keep the light illumination are received homogeneously to avoid any shadowing.
(3) Line 100 ultrasonication, please provide detailed conditions for ultrasonication (power, temperature, etc.)
(4) Why Triton X-100 was used instead of soft detergent such as DDM (dodecyl maltoside) to avoid any degradation of protein?
(5) Line 119-123 carotenoid identification was by retention time and absorption spectra. Please provide detail information how exactly the analysis was done for characterization and assignment of pigment. All carotenoid standard used must be presented in the supplementary data.
(6) Please write the detail measurement condition used for MS.

Validity of the findings

(1) The result of 1D SDS-PAAG used for ms was absent. This should be presented in a figure and the bands that will be used for ms analysis should be clearly indicated. The List of peptides from LL LH2 identified by ms should be presented well in a table (to be clearly seen as result of analysis) with indication from which band and necessary information (any code or symbols should be defined in order to be understood by broad audience)
(2) Figure 1 Absorption spectra…why the absorbance at Soret-band was truncated? Please show the spectrum until 250 nm. The y-axis label should be absorption intensity or absorbance.
(3) Table 1 Carotenoid composition…the information data is lacking. This should be supported with supplementary data consists of well-presented chromatogram profile and the full analysis of characterization and assignment of carotenoids. (There are raw data in the supplementary files, but they are very raw, and nobody can understand the meaning of the codes used and which corresponds to which). Line 119-120 "the carotenoids were identified by retention time and absorption spectra", please provide the data for the absorption spectrum of each pigments too.
(4) Figure 3, please indicate the emission wavelength monitored for the fluorescence excitation spectra. The absorption spectrum and the fluorescence excitation spectrum seem to be normalized, please indicate at which wavelength is normalized. There is a percentage value indicated on the spectrum figure, what does the value mean? How do you calculate the value? Y-axis label is presented as 1-T for the absorption spectrum, is there any reason? Why not in molar absorptivity value?

Additional comments

Figure 6-9, it is quite difficult for the audience eye to clearly see the focus of analysis. One can make transparent of those that are not necessary.
Line 244-245, percentage of carotenoid should be specified what kind of percentage.
Line 237-239, chemical formulas are not correctly written. The symbol of the number of conjugation double bond should be small n and the meaning should be indicated.

Reviewer 3 ·

Basic reporting

The English needs to be polished to be better understood by an international audience. In keeping with this, some scientific terms should conform as well to the more widely understood version (i.e. line 134 SDS-PAAG). There is also a need to go over the manuscript more carefully to correct several typos such (i.e. line 56 spectroscopy, 64 quaternary).
Lines 90–91: Need to clarify whether this new LL LH2 preparation has physicochemical properties that are modified because of the new method of purification or the LH2 itself was modified.

Experimental design

Overall, the methodology is sound and supports the objective of the paper which is to characterize new light harvesting proteins produced under low illumination conditions. One note with the samples is that it would be good to have the ‘normal or HL’ LH2 as one of the controls, considering that the “novel” LH is LL LH2. Having this control would give more insight as to how the organism adapt to low light conditions.

Validity of the findings

Although the HPLC raw data were provided, it would seem more helpful to have a figure with a direct comparison of the pigment composition for the two samples. The raw data also do not provide some key information such as the wavelength of detection which provides more context to what is observed. The authors determined that the carotenoid in the complex is rhodopin and not rhodopin glucoside. How was this determined? The ground state spectra would be the same for both pigments so mass spec of the purified pigment would confirm this but I did not see this in the paper.

Regarding Car-to-BChl a energy transfer rates as shown in Figure 3, the overlay of 1-T vs fl exc spectra shows that in the Soret region below 400 nm, the fl exc intensity is ~half that of 1-T. It is quite tricky to do these fl excitation measurements. Based on our experience, this may be due to some issues with the correction factor when measuring fluorescence in this region. Has the correction file been checked to see if it works well in the Soret or alternatively, were there control experiments done to make sure that this discrepancy does not affect the reported Car-to-BChl a energy transfer? The proposed models may explain how but what could be the reason (bigger picture) why LL LH2 has a lower efficiency than LH4? The authors also state that these LL LH2 and LH4 complexes were already reported in a previous publication and that the carotenoid composition and energy transfer efficiency did not differ significantly. How are the LH complexes reported here different/similar from those because the current data reflect some differences between the two complexes.

SDS-PAGE was supposed to have been carried out for the samples, but I did not see a figure showing the results in the paper or SI. Since this is characterization of supposedly new LH complexes I think it would be appropriate to show such data.

The discussion section could be greatly improved by expanding more on the implications of the data and results obtained and how it is a strategic tool used by the organism to survive under low light conditions.

---

## Round 0.2 · Minor Revisions

Our three reviewers have suggestions for further improvement of your manuscript. I do not expect you will have any difficulties addressing those requests.

·

Basic reporting

The article is much improved compared to its previous version, with the goals more clearly stated and corresponding better to the results and discussion.

Some details, however, remain to be corrected before publication.

1. Lines 265 to 275 should be moved from results to discussion.

2. Line 413: Principle should be principal

3. Figures 2 and 6: I apologize for what might have been an unclear comment in my previous report. What I meant is that the alignments should not be low quality pictures from Clustal website, but imported and properly annotated results. This is still not the case in the new version of the article.
For ease of read and matching the main text, in Figure 2, it would also be better to swap the denomination of the proteins in the alignment and their clarification in the legend of the figures: brief name in the alignment, and Uniprot description in the legend. The same should be done in Figure 6.

4. Figure 4 legend: It is stated that excitation spectra are represented by dotted lines, and absorption spectra by solid lines. However, both lines appear solid in the figure provided.

5. Figures 7 to 10 are still very difficult to decipher. I concur with reviewer 2 on his idea to make non-essential elements transparent or semi-transparent. If not, the authors should consider putting 2 panels rather than a superposition of both structures. The names of amino acids of interest should also be moved to blank space rather than half on the structures where they are hard to read.

Experimental design

No comment.

Validity of the findings

No comment.

·

Basic reporting

Basic reporting has shown improvement and met the expectation.

Experimental design

The revised draft has addressed the questions and made corrections.

Validity of the findings

Additional data have been added per request.
However, there is a question about Figure 1: in comparison to the previous reported findings, please look at the absorption spectra of LL LH2 and LH4 at the region 650-250 nm, the baseline increases quite dramatic, which indicating a scattering. Why? Is there any insolubilized LH particles? or Is there any degradation of your LH? Please clarify this. (enclose I am attaching the region I mentioned)

Additional comments

none

Reviewer 3 ·

Basic reporting

The edits made compared to the previous version has improved the manuscript but more extensive proofreading still needs to be done. There are still some typos within the manuscript that needs to be addressed. The manuscript will benefit greatly from a reading by a fluent English speaker or a professional editing service. In addition, having gone through the previous comments and suggestions of the reviewers, the authors reply that they have complied with the reviewer’s request but upon checking the revised document, some of these changes have not been incorporated.

e.g.
lines 166 and 172: substituted "to" should be substituted "with"

The Results and Discussion sections still need to be more clearly differentiated. Some parts of the Results section have interpretation of the data and results while the Discussion in some parts still reads a bit like literature review instead of a direct interpretation and discussion of the results.

Experimental design

Regarding the need to add the normal LH2 complex as a point of comparison, I agree that they have been extensively studied and there is plenty of literature available on them. However, since the novel LH complex being reported here is LH2, it would be informative to learn how similar/different these 2 complexes are and can help define the strategy of the organism for making such antenna complex under low light conditions. They do not have to be experimentally studied but results from the literature can be referred to.

Validity of the findings

The new figure (Supplementary file1) provided in the Supporting Information containing HPLC chromatograms is a great addition. Since the carotenoid composition is an important part of the characterization of the supposedly novel complex, I think the figure in the Supporting Information should be moved to the main text to complement the data shown in Table 3. In addition, the wavelength of detection used to generate the chromatograms should be indicated in the figure (Supplementary file1).

Other comments for the figures and tables:

It is good to have Figure 2 showing the SDS-PAGE results, what do the numbers 1-3 on the figure mean?

Figure 4: Indicate which complexes are in panels and b. Also, what do the numbers 1 and 2 correspond to? The description says fluorescence excitation (dotted lines) and absorption (solid line) but both appear as solid lines in the figure.
For the samples used for fluorescence measurements, what was the optical density especially at the excitation wavelength? This can be mentioned in the methods section.

page 31: Description of Table 2 should be for Table 3.

Table 3: anhydrorhodovibrin value for LH4- Is it 4,6 or 4.6?

Additional comments

I think the overall results and findings have significant value in the field but the manuscript still needs to be edited from its current form.

---

## Round 0.3 · accepted · Accept

Thank you for addressing the final issues mentioned by our reviewers. I am glad to accept your manuscript for publication.